# The Conundrum of Dedifferentiation in a Liposarcoma at a Peculiar Location: A Case Report and Literature Review

**DOI:** 10.3390/medicina59050967

**Published:** 2023-05-17

**Authors:** Ana-Maria Ciongariu, Adrian-Vasile Dumitru, Cătălin Cîrstoiu, Bogdan Crețu, Maria Sajin, Dana-Antonia Țăpoi, Aminia-Diana Ciobănoiu, Adrian Bejenariu, Andrei Marin, Mariana Costache

**Affiliations:** 1Pathology Department, University Emergency Hospital, 050098 Bucharest, Romania; ana-maria.ciongariu@drd.umfcd.ro (A.-M.C.); maria_sajin@yahoo.com (M.S.); dana-antonia.tapoi@drd.umfcd.ro (D.-A.Ț.); aminia-diana.ciobanoiu@rez.umfcd.ro (A.-D.C.); adrianbejenariu@yahoo.com (A.B.); andrei.marin@umfcd.ro (A.M.); m_costache_dermatopat@yahoo.com (M.C.); 2Morphology Department, Faculty of Medicine “Carol Davila”, University of Medicine and Pharmacy, 020021 Bucharest, Romania; 3Orthopedic Surgery and Traumatology Department, University Emergency Hospital, 050098 Bucharest, Romania; 4Plastic Surgery Department, Faculty of Medicine “Carol Davila”, University of Medicine and Pharmacy, 020021 Bucharest, Romania

**Keywords:** dedifferentiated, myxoid, liposarcoma, immunohistochemistry

## Abstract

Dedifferentiated liposarcoma of the deep soft tissue of the lower extremities is an infrequent finding. Myxoid liposarcoma is considered the most common soft tissue neoplasia arising in this anatomic region. Divergent differentiation usually occurs within well-differentiated liposarcoma and is exceedingly rare in a myxoid liposarcoma. We report a 32-year-old man who developed a dedifferentiated liposarcoma of the thigh on the background of a pre-existing myxoid liposarcoma. The gross examination of the surgical specimen showed a 11/7/2 cm tumour mass with solid tan-grey areas and focal myxoid degeneration. The microscopic examination revealed a malignant lipogenic proliferation, containing round cells with hyperchromatic nuclei and atypical lipoblasts, confined to the basophilic stroma with a myxoid aspect. Abrupt transition towards a hypercellular, non-lipogenic area consisting of highly pleomorphic spindle cells with atypical mitotic figures was also noted. Immunohistochemical staining was performed. Tumour cells in the lipogenic area were intensely positive for S100 and p16, and CD34 staining highlighted an arborizing capillary network. The dedifferentiated tumour areas showed positive MDM2 and CDK4 staining within neoplastic cells, with the Ki 67 proliferation marker expressed in approximately 10% of the cells. Wild-type TP53 protein expression pattern was documented. Thus, the diagnosis of a dedifferentiated liposarcoma was established. This paper aims to provide further knowledge about liposarcomas with divergent differentiation at peculiar locations, emphasizing the importance of histopathologic examination and immunohistochemical analysis for establishing the diagnosis and assessing the therapeutic response and prognosis of this condition.

## 1. Introduction

Liposarcomas are rare malignant tumours which originate from the adipose tissue and can arise at any site, with a predilection for the lower extremities and retroperitoneum [1,2]. According to the World Health Organization classification, liposarcomas develop as part of a heterogeneous group comprising atypical lipomatous tumour/well-differentiated liposarcoma, dedifferentiated liposarcoma, myxoid liposarcoma and pleomorphic liposarcoma [1,2,3]. These malignant proliferations are associated with distinct genetic abnormalities and present different clinical behaviours and metastatic potentials [1,3]. Myxoid liposarcoma is the most common adipocytic malignancy of the lower extremities, frequently affecting the proximal thigh [3,4]. Although a rare phenomenon, dedifferentiated liposarcoma and well-differentiated liposarcoma can also occur within deep somatic adipose tissue [5]. Dedifferentiated liposarcoma is an atypical lipomatous proliferation which usually develops within the retroperitoneum, but studies suggest that the incidence of non-peritoneal anatomic sites harbouring this malignancy may be underreported [5,6]. The positive and differential diagnosis of the previously mentioned lesions is crucial, especially considering the major differences between the prognosis and therapeutic management of each pathological entity [7]. In this matter, immunohistochemical analysis and genetic tests are additionally required for confirming and completing the histopathological diagnosis [7,8].

## 2. Materials and Methods

Our patient was referred to the Emergency University Hospital in Bucharest where the CT scan indicated a dense tumour mass in the adipose tissue of the left thigh, showing nodular enhancing areas. This radiology aspect was strongly suggestive of a liposarcoma. Therefore, surgical “en bloc” removal of the tumour mass was carried out, and the specimen was submitted to our pathology department. The tissue samples were fixed with 10% neutrally buffered formalin and then processed via conventional histopathological methods, using paraffin embedding, sectioning and Haematoxylin–Eosin (HE) staining. Afterwards, the sections were deparaffinised in toluene and alcohol, washed in PBS (phosphate saline buffer), incubated with normal serum, and then incubated with primary antibodies overnight. Later, washing in carbonate buffer and developing in 3,3′-diaminobenzidine hydrochloride/hydrogen peroxide and nuclear counterstain with Meyer’s Haematoxylin were performed. The immunohistochemical markers used were S100 antibody 4C4.9 mouse monoclonal, Biocare, Ki 67 antibody SP6 rabbit monoclonal, Biocare, CD 34 antibody QBEnd/10 mouse monoclonal, Biocare, p16 INK4a antibody BC42, mouse monoclonal, Biocare, p53 antibody DO7 rabbit monoclonal, Biocare, MDM 2 antibody ZR258 rabbit monoclonal, ZETA and CDK4 antibody ZR 394, rabbit monoclonal, ZETA. Furthermore, using the PubMed academic research engine, we reviewed all English language cases published until 2022. We included all cases of soft tissue tumours occurring in the lower extremities diagnosed as dedifferentiated liposarcoma in patients of any age and sex with any type of therapeutic management and clinical outcome of the disease. We excluded all patients diagnosed with other soft tissue tumours or benign proliferations of the adipose tissue. Furthermore, we excluded the cases of tumours reported as sarcomas with uncertain histotype.

## 3. Results

We present a 32-year-old male patient with no significant medical history who was referred to our hospital with a slow growing, painless tumour mass located in the upper region of the left thigh. No chemotherapy or radiation therapy, nor a history of trauma, had been documented in the patient’s medical record. A CT scan was performed, revealing a 11 cm, dense tumour mass arising in the adipose tissue of the left thigh, showing nodular enhancing areas. Moreover, multiple ipsilateral adenopathies were identified. Upon further radiology investigations, no distant metastases were detected. The patient underwent a surgical procedure consisting of wide local excision of the tumour with resection margins and lymphadenectomy. Afterwards, the surgical specimens were submitted to our Pathology Department.

The gross examination revealed a 11/7/2 cm well-circumscribed, multinodular tumour with a heterogeneous aspect on the cut surface. The lesion presented a course lobular contour and yellow-tan colour, firm consistency and small gelatinous areas, as well as focal nodular, solid zones.

The histological examination showed a malignant mesenchymal proliferation with a lobular growth pattern, exhibiting round–oval and fusiform neoplastic cells with hyperchromatic nuclei and occasional nuclear pseudoinclusions, separated by a loose, lightly basophilic stroma, containing extracellular mucoid pools (Figure 1 and Figure 2).

Moreover, a prominent vasculature, formed by delicate arborizing capillaries with a plexiform arrangement, was identified. The tumour lobules were surrounded by a rim of atypical lipoblasts. The malignant tumour proliferation displayed abrupt transition towards a divergent component (Figure 3) consisting of highly pleomorphic tumour cells with atypical mitotic figures, with associating small necrotic and haemorrhagic areas (Figure 4, Figure 5 and Figure 6).

No lympho-vascular tumour emboli or perineural invasion were detected. Four surgically excised inguinal lymph nodes were examined, with mild reactive histiocytosis and no tumour invasion. As the histopathologic features of the lesion was highly suggestive of a dedifferentiated liposarcoma, several additional ancillary tests were carried out. First, immunohistochemical analysis was used to determine the following tumour immunophenotype: neoplastic cells of the lipogenic tumour component presented intense S100 and p16 protein expression (Figure 7 and Figure 8), while CD 34 staining underlined the presence of plexiform arborizing capillary blood vessels within the lipogenic tumour component (Figure 9).

Both components of the divergent tumour proliferation were associated with a wild type TP53 expression pattern, and approximately 10% of the tumour cells expressed the Ki 67 proliferation marker (Figure 10).

The expression of MDM2 and CDK 4 markers within the tumour cells of the dedifferentiated component eventually established the diagnosis of a dedifferentiated liposarcoma developed on the background of a myxoid liposarcoma (Figure 11 and Figure 12).

The patient did not present with any metastatic lesions at the time of diagnosis. Therefore, the tumour proliferation was categorized as a stage III A lesion (pT2 pN0 M0) according to the TNM evaluation system. Genetic testing was also recommended to confirm and complete the diagnosis. The patient was later referred to the oncology department for evaluation and initiation of an adequate therapeutic scheme. After a six-month follow-up, the patient is disease-free, with no evidence of recurrence or metastases.

## 4. Discussion

Dedifferentiated liposarcoma is a malignant proliferation of the adipose tissue which typically involves the retroperitoneal space and, infrequently, lower extremities, spermatic cord and structures of the head and neck [8,9]. Dedifferentiated liposarcoma of the thigh is an uncommon entity, as this anatomic region is prone to develop myxoid liposarcoma or non-adipogenic malignant mesenchymal proliferations [9,10]. Although there are a few cases reported, dedifferentiated liposarcoma should be taken into consideration when examining soft tissue tumour masses of the lower extremities [11]. Myxoid liposarcoma is one of the most frequent soft tissue sarcomas and is associated with peculiar genetic mutations, implying an unfavourable prognosis [10,12]. Currently, the various liposarcoma subtypes are recognised to have different biological behaviour in correlation with their distinct genetic alteration [13,14]. The pathogenesis of dedifferentiated liposarcoma can be summarized by the presence of a high-level amplification in the chromosomal 12q13–15 region involving the CDK4 and MDM2 cell cycle oncogenes [5,15]. The genetic abnormalities are a common feature of dedifferentiated liposarcoma and atypical lipomatous tumour/well-differentiated liposarcoma [5,15]. However, recent advances in next generation sequencing show that novel genes and pathways occur in dedifferentiated liposarcoma [16]. Whole genome sequencing disclosed various genetic alterations, including 6q23 and 1q32 coamplification, which may be harboured by dedifferentiated liposarcoma [15,16]. Considering the rarity of this malignant tumour, anatomical location and histotype are infrequently mentioned throughout the results of the surveys, with no specific data upon liposarcoma of the extremities available [11,16]. Although the role of MDM2 and CDK4 genes in the pathogenesis of dedifferentiated liposarcoma is well understood, the most recent studies have demonstrated that DDIT3 amplification is also identified in patients developing this malignancy [17]. As it was mentioned, due to the uncommonness of this soft tissue neoplasm, the correlations between DDIT3 amplification and the tumour’s location and clinical behaviour are currently scarce [17]. Moreover, numerous studies on the pathogenesis of myxoid liposarcoma have also been carried out, considering the peculiarities of this entity’s prognosis and therapeutic management. The FUS: DDIT3 fusion oncoprotein is the current genetic hallmark of myxoid liposarcoma, and it is considered that it acts like an aberrant transcription factor [12,18]. Furthermore, studies reveal that overexpression of DDIT3 along with MDM2 and CDK4 is also identified in dedifferentiated liposarcoma, although considered a genetic alteration strongly associated with myxoid liposarcoma [12]. In addition, studies suggest that the aforementioned genetic abnormality may determine the presence of myxoid-like histopathologic aspects within dedifferentiated liposarcoma [12]. As a result, thorough correlations between the morphologic and immunohistochemical features and the genetic background of these particular soft tissue neoplasms are required for achieving the correct diagnosis.

Regarding the challenging histopathologic features of some liposarcomas which may lead to a misinterpreted diagnosis, the question of heterologous differentiation and tumour metaplasia has also been put [19] Heterologous differentiation within a myxoid liposarcoma has only rarely been reported [20,21]. The main discussion in this matter refers to the morphology of the proliferated heterologous elements which may highly resemble tumour metaplasia [19]. Up to date, five cases of myxoid liposarcoma with cartilaginous differentiation confirmed on cytogenetic analysis have been reported [20]. In our case, the proliferation displayed an abrupt transition towards a non-lipogenic area composed of pleomorphic spindle cells with unequivocal malignant features, ruling out metaplasia. Moreover, dedifferentiated liposarcoma exhibiting a prominent myxoid stroma is an important histopathological finding which should be taken into consideration when analysing malignant lesions with adipocytic differentiation [22,23]. However, this uncommon histopathologic aspect has mainly been identified during the examination of tumour masses involving the retroperitoneal space and paratesticular soft tissue [22,23]. In our case, microscopic examination revealed the presence of basophilic matrix enclosing sarcomatous neoplastic cells and containing a myxoid liposarcoma-like plexiform capillary vasculature, adjacent to a non-lipogenic area with divergent differentiation. Studies provide evidence of prominent vascular capillary networks within dedifferentiated liposarcoma with myxoid stroma and suggest both myxoid liposarcoma and myxofibrosarcoma should be taken into consideration as a differential diagnosis [22,23,24]. Apart from the differential diagnosis between myxoid liposarcoma and its dedifferentiated counterpart, there is a small focus on highly pleomorphic liposarcomas with mixed elements [25]. Campbell et al. report a case of a mixed lesion of the axilla comprising well-differentiated, dedifferentiated elements adjacent to a myxoid liposarcoma component [25]. In relation to this phenomenon, we highlight the importance of immunohistochemical tests and genetic analysis as reliable methods for assessing the response to neoadjuvant therapy [26].

The evaluation of the tumour immunophenotype can facilitate the differential diagnosis of lesions with similar histomorphology, and it is also relevant in predicting the therapeutic tumour response [27]. First, immunohistochemical analysis can be used to distinguish dedifferentiated liposarcoma from other types of malignant mesenchymal proliferations [8]. Currently, this diagnostic procedure is guided by the use of MDM2 and CDK4 markers and confirmed using additional molecular testing of the corresponding genes amplification [8,28]. Ancillary tests are necessary when dealing with tumours displaying histopathologic features suggestive of a dedifferentiated or well-differentiated liposarcoma, but with no detectable lipogenic component [28]. The studies that were carried out performed MDM2, CDK4 and p16 staining on samples originating from tumours with the aforementioned histotype. The highest sensitivity and specificity were noted for the MDM2 marker, as an excellent correlation was obtained between the results of MDM2 fluorescence and in situ hybridization [28,29]. The previous immunomarker was followed by CDK 4 and p16 which also provide sensitive and specific staining for dedifferentiated liposarcoma [7,27]. Although there are a few reported cases of dedifferentiated liposarcoma of the thigh, MDM2, CDK4 and p16 were used as part of a panel to confirm the diagnosis [30,31,32]. On the other hand, immunohistochemical study of myxoid liposarcoma is currently of limited use. As it was earlier mentioned, myxoid liposarcoma is associated with recurrent molecular alterations implying DDIT3 rearrangement; therefore, anti-DDIT3 immunoreactivity is the only specific staining for this tumour proliferation [25,32]. As this immunomarker is not yet available in regular antibody panels, the ancillary study of myxoid liposarcoma of any location is limited to the examination of S100 and p16 expression and of the status of the TP53 mutation [33,34,35]. S100 expression within neoplastic cells can be used to differentiate myxoid liposarcoma from a non-adipocytic neoplasm with myxoid stroma, while p16 has proved to be practical in distinguishing the tumour in question from its benign counterparts [34,36]. Although the expression of TP53 within lesions from the pathologic spectrum of liposarcoma has been widely examined, researches do not provide evidence of a major significance of this mutation for the overall prognosis of the neoplastic disease [37,38]. In our patient, the applied immunohistochemical stains provided a precise recognition of the divergent differentiation. As dedifferentiated liposarcoma of the thigh is an uncommon finding, performing ancillary tests with the tissue samples is mandatory for establishing the diagnosis.

Clinical behaviour, prognosis and therapeutic management of soft tissue liposarcoma has been analysed in several studies for a notably long time. To this moment, the prognosis of patients developing dedifferentiated liposarcoma of the extremities can be estimated considering multiple prognostic factors. First, dedifferentiated liposarcoma is regarded as a malignant tumour with a better prognosis compared to myxoid and pleomorphic liposarcoma and other soft tissue sarcomas, exploiting the fact that the majority of these adipocytic tumours harbour MDM2 and CDK4 expression [6,7]. Clinical trials support the benefit of CDK4 inhibitor abemaciclib for the treatment of dedifferentiated liposarcoma with CDK4 and MDM2 expression and strongly recommend the further development of anti-MDM2 agents, which, at the moment, have limited clinical use due to bone marrow toxicity resulting in thrombocytopenia [6,7]. In addition, trials using immune checkpoint inhibitors as a targeted therapy for dedifferentiated liposarcomas have been proposed, as PD-1 expression has been identified within liposarcoma [39,40]. Although there are several on-going trials investigating new targeted therapies for liposarcoma, patients diagnosed with this neoplasm currently receive classic cytotoxic chemotherapy agents and radiotherapy [7,15]. The most widely used therapeutic schemes include doxorubicin, docetaxel and gemcitabine, which are prescribed as part of adjuvant treatment [41].

Immunohistochemical and genetic analyses of dedifferentiated liposarcoma and the other malignant tumours with adipocytic differentiation represent an important research purpose with important new perspectives. Surveys implying new correlations between molecular biology data, genetic analysis and immunohistochemistry have constantly been performed, in order to discover advanced tools for diagnosis and customized oncological treatment. For example, there is an interest in investigating the expression of GATA proteins and other molecules within various malignant tumours. Transcription factors from the GATA protein family are involved in adipose tissue development and maturation [41]. Their corresponding immunomarkers may be expressed within fat cells precursors and may facilitate the differential diagnosis of morphologically similar lesions [42]. CD 70-CD27 ligand–receptor complex stimulates the development of T lymphocytes [43]. At the moment, there is evidence of CD 70 expression in lymphomas and several sarcomas, with a significant relevance in predicting tumour response to therapy with specific inhibitory agents [43].

## 5. Conclusions

Dedifferentiation within a myxoid liposarcoma is an exceedingly rare phenomenon, although cases of myxoid liposarcoma with heterologous differentiation, as well as dedifferentiated liposarcoma with a myxoid stroma have been reported so far. The precise diagnosis of dedifferentiated liposarcoma requires the use of ancillary studies, implying immunohistochemistry and genetic analysis. Differentiating the aforementioned malignant tumour from other soft tissue sarcomas is crucial, considering the significant differences regarding the prognosis and treatment of each entity. Therapeutic procedures have constantly been improved. Therefore, a diligent assessment of the tumour histotype, immunophenotype and genetic background is mandatory for the successful management of patients developing dedifferentiated liposarcomas.

## Figures and Tables

**Figure 1 medicina-59-00967-f001:**
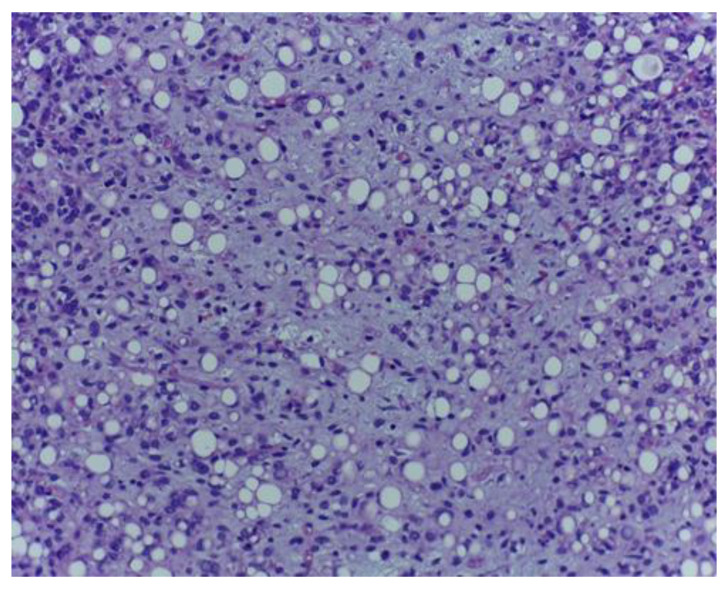
Lipogenic tumour component comprising atypical lipoblasts encompassed by a lightly basophilic matrix with myxoid aspect. H.E., ob. 200×.

**Figure 2 medicina-59-00967-f002:**
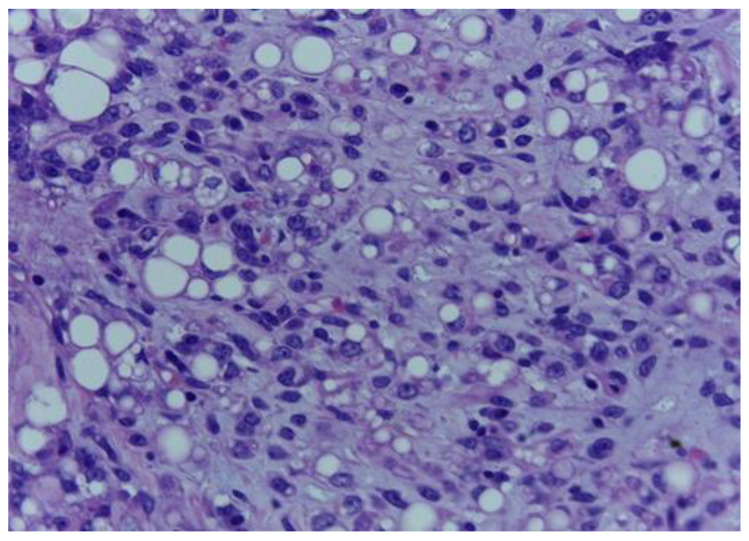
The tumour proliferation with adipocytic differentiation displays round and spindle neoplastic cells with hyperchromatic nuclei. H.E., ob. 400×.

**Figure 3 medicina-59-00967-f003:**
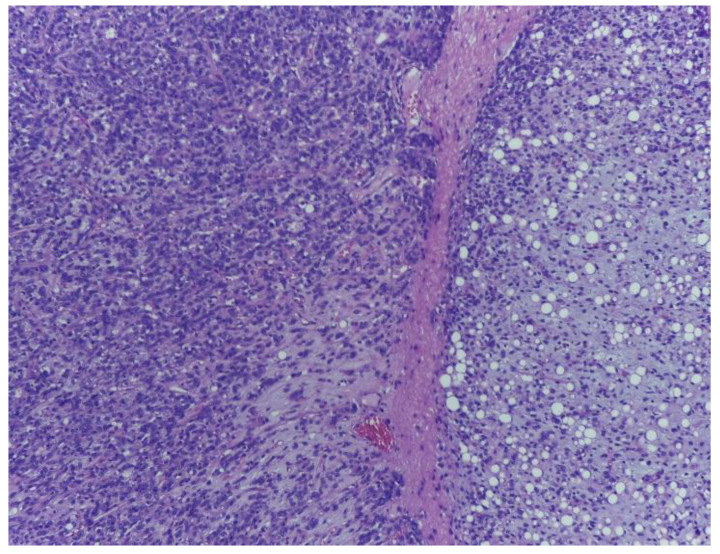
Malignant tumour proliferation with adipocytic differentiation exhibiting abrupt transition towards a non-lipogenic area, containing atypical spindle cells with fasciculate arrangement. H.E., ob. 100×.

**Figure 4 medicina-59-00967-f004:**
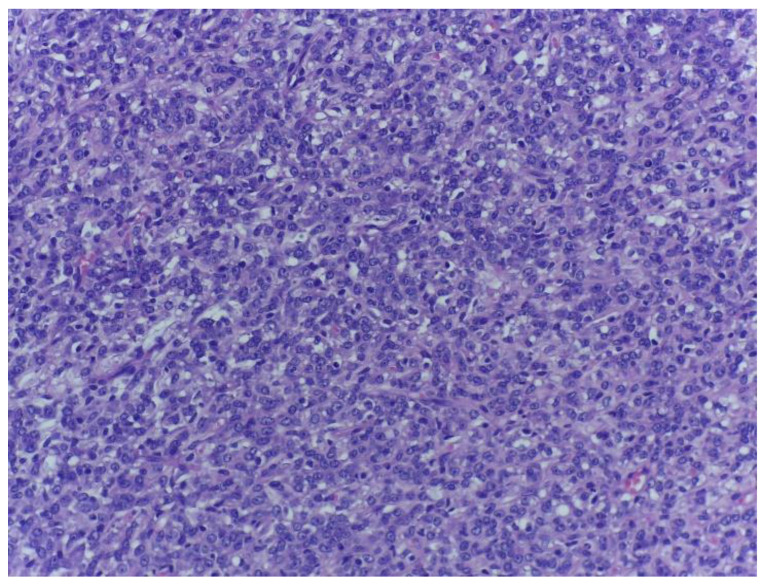
Dedifferentiated tumour area composed of spindle cells with hyperchromatic, moderately pleomorphic nuclei, showing no lipogenic areas and inapparent lipoblasts. H.E., ob. 200×.

**Figure 5 medicina-59-00967-f005:**
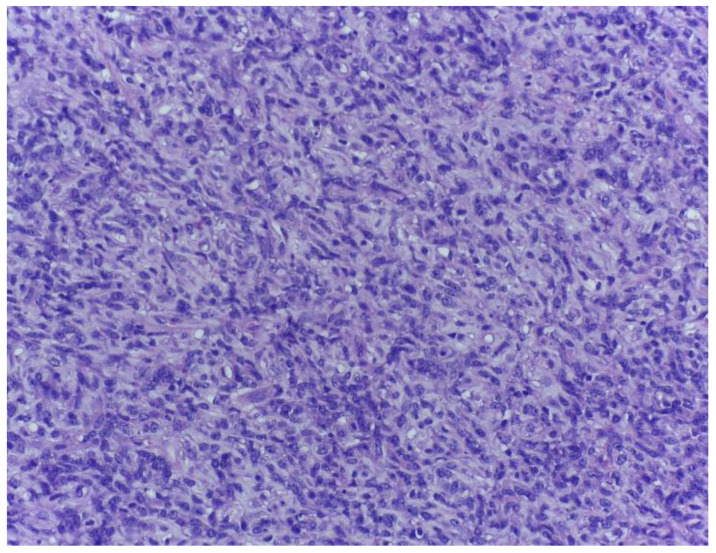
Dedifferentiated tumour component exhibiting intersecting fascicles of malignant spindle cells with no lipogenic areas. H.E., ob. 200×.

**Figure 6 medicina-59-00967-f006:**
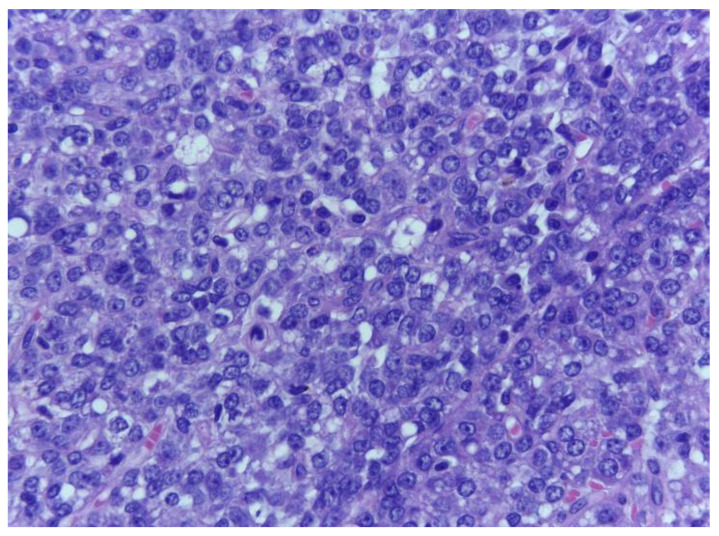
The non-lipogenic tumour component displays spindle cells with hyperchromatic nuclei; some of them with conspicuous nucleoli, as well as mitotic figures. HE, ob. 400×.

**Figure 7 medicina-59-00967-f007:**
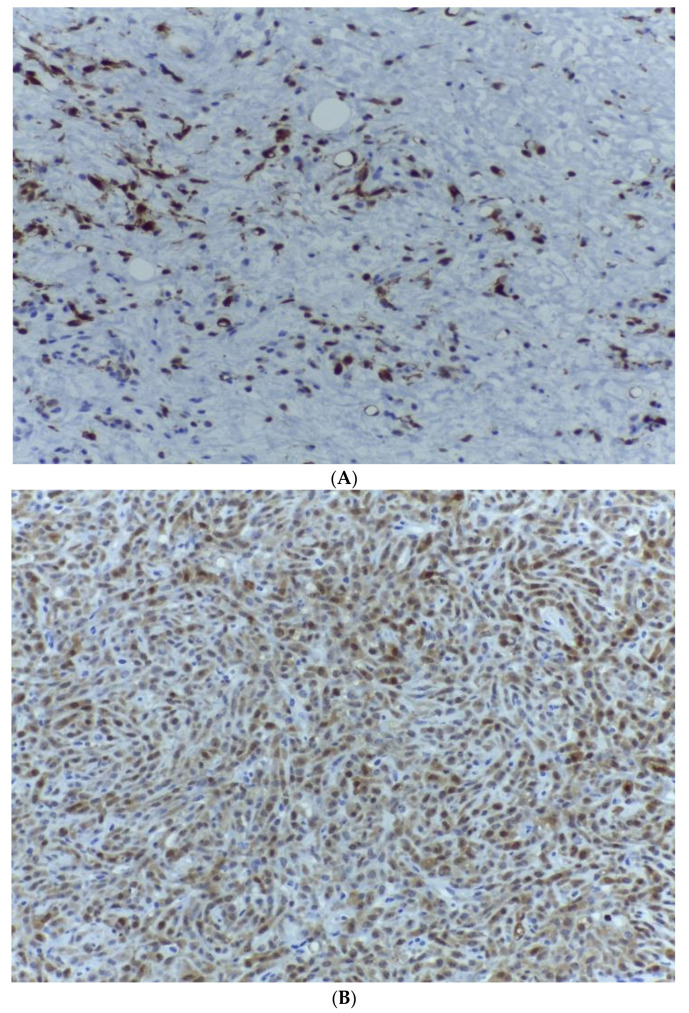
(**A**) S100 expression within the malignant cells of the lipogenic tumour area with myxoid stroma. S100, ob. 200×. (**B**) S100 expression within the malignant cells of the dedifferentiated tumour component. S100, ob. 200×.

**Figure 8 medicina-59-00967-f008:**
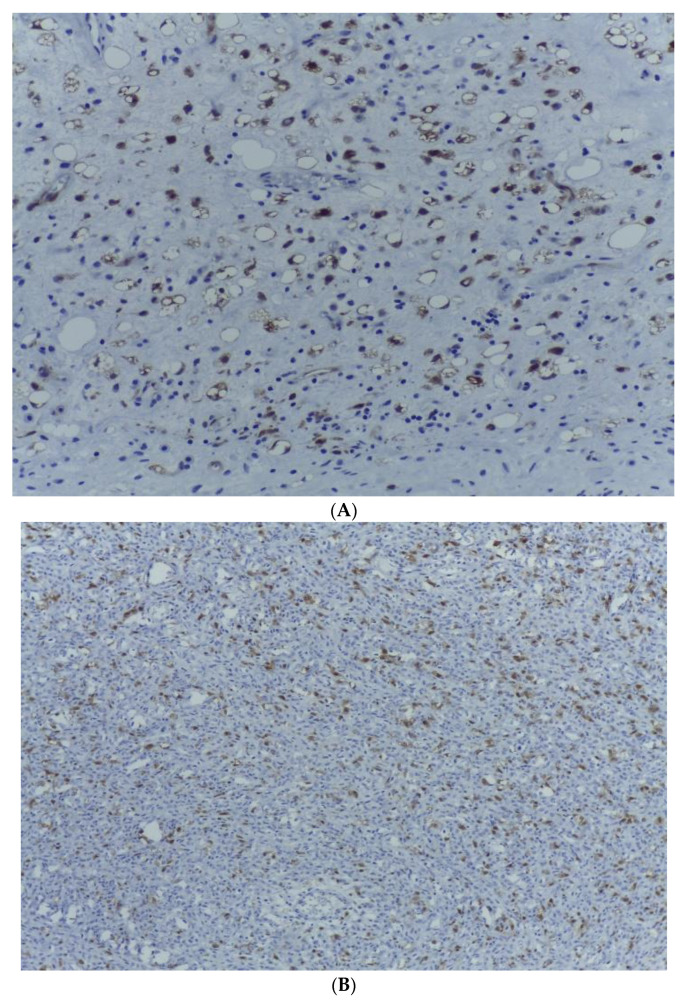
(**A**) Strong, diffuse p16 expression within the malignant cells of the lipogenic tumour area. p16, ob. 200×. (**B**) Strong, diffuse p16 expression within the malignant cells of the non-lipogenic tumour area. p16, ob. 200×.

**Figure 9 medicina-59-00967-f009:**
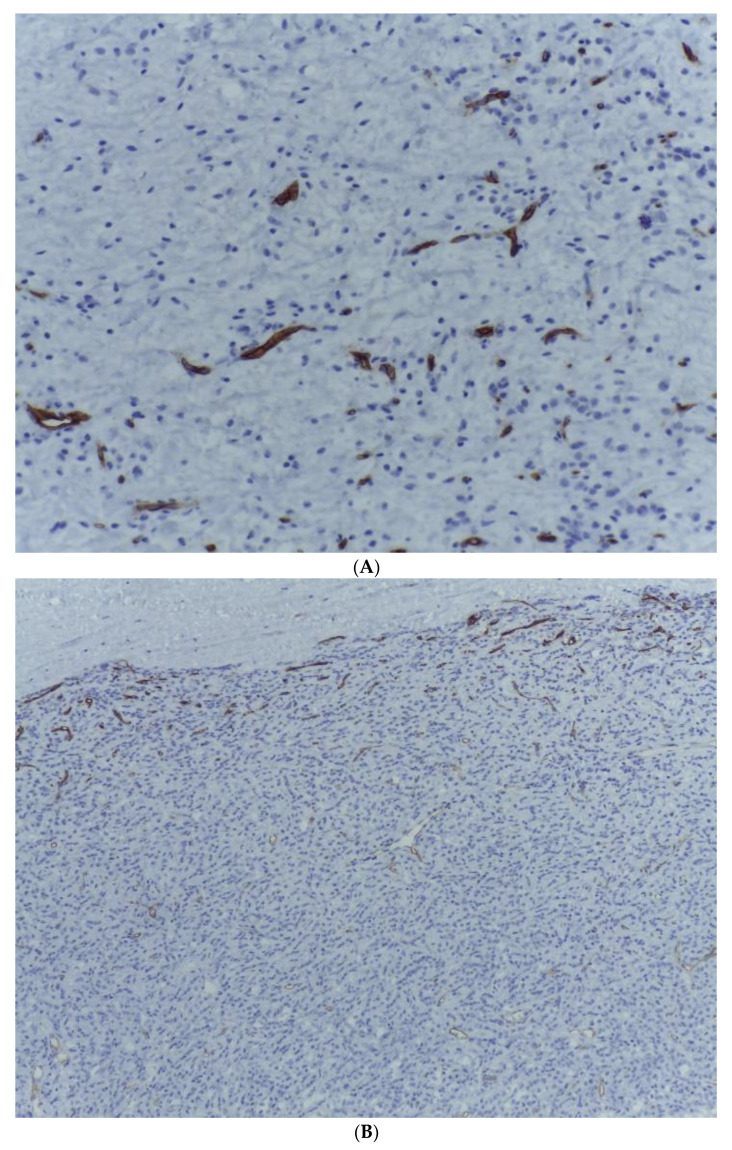
(**A**) CD34 staining highlighting the prominent “chicken-wire” capillary network within the lipogenic tumour area. HE, ob. 200×. (**B**) Tumour cells within the dedifferentiated tumour component are negative for CD 34. Immunohistochemical staining also demonstrates a delicate arborizing vasculature present within the outer limits of the proliferation. CD 34, ob. 100×.

**Figure 10 medicina-59-00967-f010:**
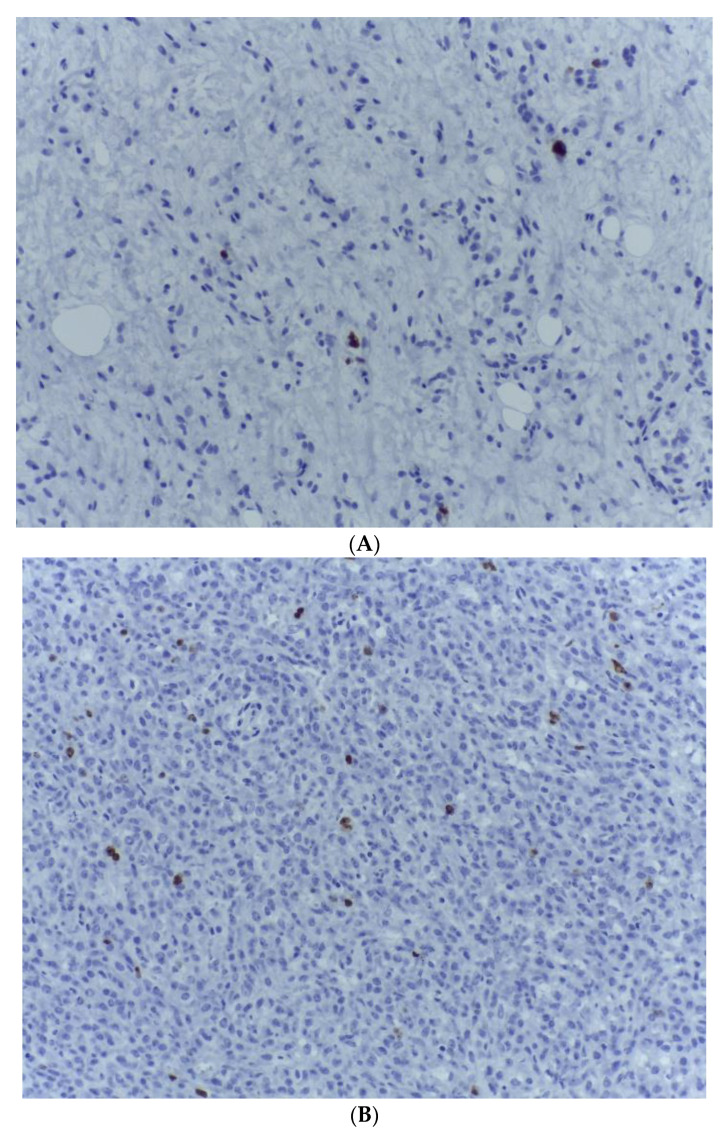
(**A**) The Ki 67 proliferation marker is expressed within around 10% of the malignant cells from the lipogenic area with myxoid stroma. Ki 67, ob. 200×. (**B**) The Ki 67 proliferation marker is expressed within 10% of the malignant spindle cells from the dedifferentiated tumour area. Ki 67, ob. 200×.

**Figure 11 medicina-59-00967-f011:**
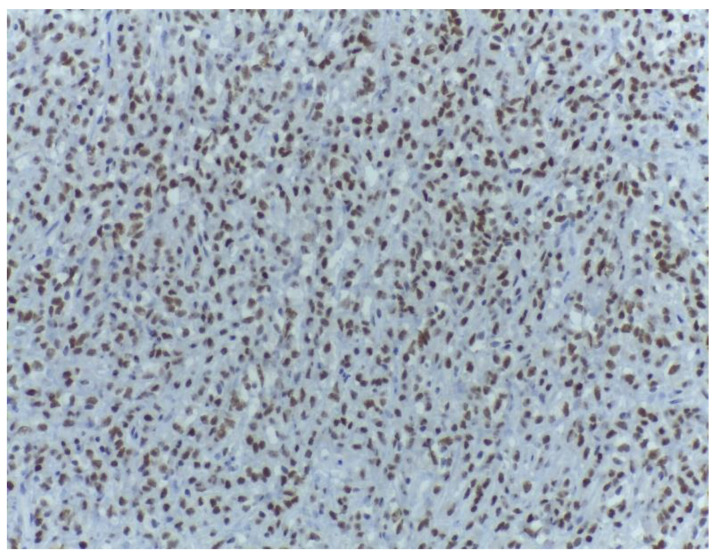
Strong MDM2 expression within the malignant cells of the non-lipogenic tumour area. MDM2, ob. 200×.

**Figure 12 medicina-59-00967-f012:**
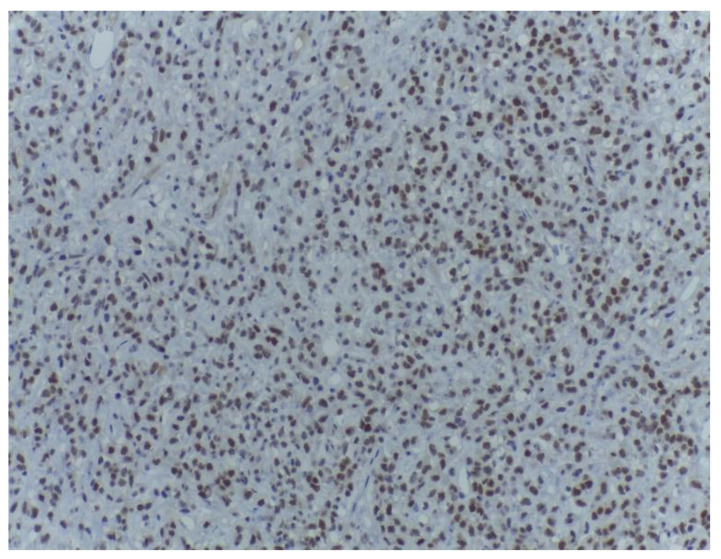
Strong CDK4 expression within the malignant cells of the non-lipogenic tumour area. CDK4, ob. 200×.

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
