# Peer review of "The Conundrum of Dedifferentiation in a Liposarcoma at a Peculiar Location: A Case Report and Literature Review"

_medicina, 2023, doi:10.3390/medicina59050967_

Round 1

Reviewer 1 Report

The manuscript from Ciongariu and colleagues report an un common lesion of dedifferentiated liposarcoma (DDLPS) of the thigh occurring from a preexisting myxoid liposarcoma (MLPS).

The paper is interesting since it report an less reported sarcoma occurrence.

The manuscript will benefit from the followings

 1.       The authors should include CT scan images of the diagnosed DDLPS.

2.       It should be useful to have more details about MLPS background lesion (FUS-CHOP analysis and so on).

3.       Some relevant references are missing. The following papers should be included in the manuscript: “Current classification, treatment options, and new perspectives in the management of adipocytic sarcomas”.

Minor revisions are requested

Author Response

Thank you for your valuable opinion. Please see the attachment. 

Reviewer 2 Report

The authors present a case report this fact must be clearly declare in the head line of aricle. 
the authors describe a large immunohistochemistry of diagnose sarcoma with a hughe number of histology photos. No any of them have any original analyse for sarcoma classification or pathoogy resonable interpretation in diagnostic set. 
the form of the article is near to the journals focused on histopthology only. 
the clinical treatment and treatmen modality used by this case are not describe no. Linical data TNM classification treatment protocl were not described clearly. No surviving data were described.

in the disscusion is described the presence of PD-L1 but not relevant data for this case was péresented. 

Author Response

(The authors gave the same response as above.)
